



# Residual Linewidth in Magic-Angle Spinning Solid-State NMR

Matías Chávez[1], Thomas Wiegand[1], Alexander A. Malär[1], Beat H. Meier[1], and Matthias Ernst[1]

[1]Physical Chemistry, ETH Zürich, Vladimir-Prelog-Weg 2, 8093 Zürich, Switzerland

**Correspondence:** Matthias Ernst (maer@ethz.ch), Beat H. Meier (beme@ethz.ch)

**Abstract.** Magic-angle spinning is routinely used to average anisotropic interactions in solid-state NMR. Due to the fact, that the Hamiltonian of a strongly-coupled spin system does not commute with itself at different time points during the rotation, second-order and higher-order terms lead to a residual line broadening in the observed resonances. Additional truncation of the residual broadening due to isotropic chemical-shift differences can be observed. We analyze the residual line broadening in coupled proton spin systems based on theoretical calculations of effective Hamiltonians up to third order using Floquet theory and compare these results to numerically obtained effective Hamiltonians in small spin systems. We show that at spinning frequencies beyond 50 kHz, second-order terms dominate the residual line width leading to a $1/\omega_r$ dependence of the second moment which we use to characterize the line width. However, chemical-shift truncation leads to a partial $\omega_r^{-2}$ dependence of the line width which looks as if third-order effective Hamiltonian terms are contributing significantly. We show that second-order contributions not only broaden the line but also lead to a shift of the center of gravity of the line. Experimental data reveals such spinning-frequency dependent line shifts in proton spectra in model substances that can be explained by line shifts induced by the second-order dipolar Hamiltonian.

## 1 Introduction

Magic-angle spinning (MAS) (Andrew et al., 1958, 1959; Lowe, 1959) is a prerequisite for almost all high-resolution solid-state NMR spectroscopic techniques. Over the years, a steady increase in MAS frequencies has been achieved (Böckmann et al., 2015) up to 170 kHz for cylindrical rotors (Lin et al., 2018; Samoson, 2019; Schledorn et al., 2020) and also alternate rotor designs have been implemented based on spheres (Chen et al., 2018; Gao et al., 2019). With the spinning frequencies currently available, proton-detection experiments in fully protonated and labelled proteins have become feasible (Andreas et al., 2016; Agarwal et al., 2014; Stöppler et al., 2018; Medeiros-Silva et al., 2016; Vasa et al., 2019; Stanek et al., 2016; Nishiyama et al., 2014; Struppe et al., 2017; Schubeis et al., 2018; Vasa et al., 2018). The residual line width of such proton-detected protein spectra is in the order of a few hundred Hertz (full width at half maximum (FWHM)) and decreases with increasing spinning frequency, $\omega_r$. There are different contributions to the residual line width as discussed in detail in (Penzel et al., 2019) which can be classified as homogeneous or inhomogeneous contributions according to Maricq and Waugh (1979). The





inhomogeneous contributions can be refocused in a Hahn-echo experiment (Hahn, 1950), e.g., magnetic-field inhomogeneity
or susceptibility broadening, while homogeneous contributions cannot be refocused. The homogeneous contributions originate
either from coherent terms that are due to incomplete averaging by MAS or from incoherent relaxation due to stochastic
dynamic processes in the molecule. The incoherent relaxation contribution is expected to change only little with a change
in MAS frequency since the only difference comes from the spinning-frequency dependent sampling of the spectral-density
function $J(\omega_{\mathrm{r}})$ (Schanda and Ernst, 2016) that contributes to $T_2$. The refocused line width FWHM(hom) $= \pi/T_2'$, is, therefore,
the sum of a nearly constant term $\pi/T_2$ and the coherent contribution that scales with the spinning frequency.

To describe experiments with time-dependent Hamiltonians like it is the case under MAS, average-Hamiltonian theory
(AHT) (Haeberlen and Waugh, 1968; Haeberlen, 1976; Ernst et al., 1990) or Floquet theory (Scholz et al., 2010; Leskes et al.,
2010) can be used to calculate effective time-independent Hamiltonians to different orders. Since the dipolar coupling is a
traceless second-rank tensor, one would expect no contribution in the first-order term. However, second-order (commutator)
terms and third-order (double-commutator) terms are possible as well as higher-order contributions. The second-order terms are
expected to scale with $1/\omega_{\mathrm{r}}$ while in general terms of order $n$ are expected to scale with $\omega_{\mathrm{r}}^{-(n-1)}$. Experimental observations
of the residual homogeneous line width as a function of spinning frequency show that it can often be approximated by a linear
correlation with the inverse of the spinning frequency with some deviation that indicate a partial inverse quadratic dependence
(Nishiyama, 2016; Sternberg et al., 2018; Penzel et al., 2019; Schledorn et al., 2020). This has been attributed to third-order
contributions to the effective Hamiltonian or to chemical-shift effects (Sternberg et al., 2018; Moutzouri et al., 2020).

Second-order effective Hamiltonians under MAS for strongly-coupled spin systems have been calculated before based on
AHT (Brunner et al., 1990b, a; Brunner, 1993, 2001; Malär et al., 2019) for three-spin sub systems with an arbitrary geometry.
We extend this work to a general solution for the third-order terms based on Floquet theory. The analytical solutions can be used
to calculate spectra numerically based on different orders of the analytical solution and analyze their scaling behavior under
MAS. Alternatively, exact effective Hamiltonians based on the numerical propagator (calculated by time slicing of the time-
dependent Hamiltonian) can be calculated using the matrix logarithm of the propagator (Liu et al., 1990). When calculating an
effective Hamiltonian from the propagator using the matrix logarithm, one has to keep in mind that the eigenvalues can only
be determined modulo the inverse of the cycle time which is in our case the spinning frequency. For fast spinning, i.e. short
cycle times, this is usually not a problem since the eigenvalues of the effective Hamiltonian are within the interval $\pm\omega_{\mathrm{r}}/2$.
Based on these different effective Hamiltonians, we calculate the second moment ($M_2$) of the line which can be correlated to
an equivalent line width of a Gaussian line (Mehring, 1983).

We show that third-order terms do not play a critical role in the residual line width at MAS spinning frequencies beyond
50 kHz. Due to the structure of the second-order Hamiltonian, the lines are not only broadened but also shifted which can
be characterized by the first moment ($M_1$) of the line which describes the center of gravity of the line. These line shifts can
be observed experimentally. The experimentally observed deviation from the expected $1/\omega_{\mathrm{r}}$ dependence can be reproduced
in numerical calculations only if chemical-shift differences are considered. The additional truncation of the residual dipolar
couplings by the chemical shift is the reason for this difference.





## 2 Theory

We assume a homonuclear spin system with chemical shifts and homonuclear dipolar couplings under MAS. The time-dependent Hamiltonian for such a system can be written as

$$\hat{\mathcal{H}}(t) = \sum_{p=1}^{N} \omega_p \hat{I}_{pz} + \sum_{p=1}^{N-1}\sum_{q=p+1}^{N} \omega_{pq}(t)\left[2\hat{I}_{pz}\hat{I}_{qz} - \left(\hat{I}_{px}\hat{I}_{qx} + \hat{I}_{py}\hat{I}_{qy}\right)\right] \tag{1}$$

where the time-dependent dipolar coupling is given by

$$\omega_{pq}(t) = \sum_{m=-2}^{2} \omega_{pq}^{(m)} e^{im\omega_{\mathrm{r}}t} \tag{2}$$

and the Fourier coefficients are defined by

$$\omega_{pq}^{(m)} = \frac{1}{\sqrt{6}} \sum_{m=2}^{2}\sum_{m'=-2}^{2} d_{m,0}^2(-\theta_{\mathrm{m}}) e^{-im\gamma} d_{m',m}^2(\beta) e^{-im'\alpha} e^{-im'\phi_{pq,12}} d_{0,m'}^2(\theta_{pq,12})\sqrt{\frac{3}{2}}\delta_{pq} \tag{3}$$

Here, $\delta_{pq}$ is the anisotropy of the dipolar coupling and $d_{m,m'}^l(\theta)$ are the reduced Wigner rotation matrix elements. We transform the dipolar-coupling tensor from the principal-axes system (PAS) to the laboratory-frame system (LAB) by three consecutive Euler rotations: (i) a rotation by $(0,\theta_{(pq)},\phi_{(pq)})$ from the PAS of spin pair $(pq)$ to the PAS of spin pair $(12)$; (ii) a rotation by 70 $(\alpha,\beta,\gamma)$ from the PAS of spin pair $(12)$ to the rotor fixed frame; (iii) a rotation by $(-\omega_{\mathrm{r}}t, -\theta_{\mathrm{m}}, 0)$ from the rotor-fixed frame to the laboratory-frame of reference. The Hamiltonian is periodic with a single frequency and can be written as a Fourier series

$$\hat{\mathcal{H}}(t) = \sum_{n=-2}^{2} \hat{\mathcal{H}}^{(n)} e^{in\omega_{\mathrm{r}}t} \tag{4}$$

and the Fourier coefficients of the Hamiltonian are given by

$$\hat{\mathcal{H}}^{(0)} = \sum_{p=1}^{N} \omega_p \hat{I}_{pz} \tag{5}$$


$$\hat{\mathcal{H}}^{(n)} = \sum_{p=1}^{N-1}\sum_{q=p+1}^{N} \omega_{pq}^{(n)}\left[2\hat{I}_{pz}\hat{I}_{qz} - \left(\hat{I}_{px}\hat{I}_{qx} + \hat{I}_{py}\hat{I}_{qy}\right)\right] \tag{6}$$

Based on single-mode Floquet theory we can now calculate the first three orders of the effective Hamiltonian (Scholz et al., 2010; Hellwagner et al., 2020) according to

$$\begin{aligned}
\bar{\hat{\mathcal{H}}} &= \bar{\hat{\mathcal{H}}}^{(1)} + \bar{\hat{\mathcal{H}}}^{(2)} + \bar{\hat{\mathcal{H}}}^{(3)} + \cdots \\
\quad &= \hat{\mathcal{H}}^{(0)} + \frac{1}{2}\sum_{n\neq 0}\frac{\left[\hat{\mathcal{H}}^{(n)}, \hat{\mathcal{H}}^{(-n)}\right]}{n\omega_{\mathrm{r}}} + \frac{1}{2}\sum_{n\neq 0}\frac{\left[\left[\hat{\mathcal{H}}^{(n)}, \hat{\mathcal{H}}^{(0)}\right], \hat{\mathcal{H}}^{(-n)}\right]}{(n\omega_{\mathrm{r}})^2} + \frac{1}{3}\sum_{k,n\neq 0}\frac{\left[\hat{\mathcal{H}}^{(n)}, \left[\hat{\mathcal{H}}^{(k)}, \hat{\mathcal{H}}^{(-k-n)}\right]\right]}{n\omega_{\mathrm{r}}k\omega_{\mathrm{r}}} + \cdots
\end{aligned} \tag{7}$$





Second-order terms are fully described by a three-spin system while third-order terms require a four-spin system to obtain all possible terms. The first three orders of the effective Hamiltonian for a dipolar-coupled spin system are given by

$$\hat{\bar{\mathcal{H}}}^{(1)} = \hat{\mathcal{H}}^{(0)} = \sum_{p=1}^{N} \omega_p \hat{I}_{pz} \tag{8}$$

$$\hat{\bar{\mathcal{H}}}^{(2)} = \frac{1}{2} \sum_{n \neq 0} \frac{\left[\hat{\mathcal{H}}^{(n)}, \hat{\mathcal{H}}^{(-n)}\right]}{n\omega_{\mathrm{r}}} = \sum_{p,q,r} \omega_{pqr}^{(\mathrm{eff})} \hat{I}_{pz} \left( \hat{I}_q^+ \hat{I}_r^- - \hat{I}_q^- \hat{I}_r^+ \right) \tag{9}$$


$$\begin{aligned} \hat{\bar{\mathcal{H}}}^{(3)} =& \frac{1}{2} \sum_{n \neq 0} \frac{\left[\left[\hat{\mathcal{H}}^{(n)}, \hat{\mathcal{H}}^{(0)}\right], \hat{\mathcal{H}}^{(-n)}\right]}{(n\omega_{\mathrm{r}})^2} + \frac{1}{3} \sum_{k,n \neq 0} \frac{\left[\hat{\mathcal{H}}^{(n)}, \left[\hat{\mathcal{H}}^{(k)}, \hat{\mathcal{H}}^{(-k-n)}\right]\right]}{n\omega_{\mathrm{r}} k\omega_{\mathrm{r}}} \\ =& \sum_p \omega_p^{(\mathrm{eff})} \hat{I}_{pz} + \sum_{p,q} \omega_{pq,z}^{(\mathrm{eff})} \left(2\hat{I}_{pz}\hat{I}_{qz}\right) - \omega_{pq}^{(\mathrm{eff})} \left(\hat{I}_{px}\hat{I}_{qx} + \hat{I}_{py}\hat{I}_{qy}\right) + \sum_{p,q,r} \omega_{pqr}^{(\mathrm{eff})} \hat{I}_{pz} \left(\hat{I}_q^+ \hat{I}_r^- + \hat{I}_q^- \hat{I}_r^+\right) \\ &+ \sum_{p,q,r,s} \omega_{pqrs,z}^{(\mathrm{eff})} \hat{I}_{pz}\hat{I}_{qz} \left(\hat{I}_r^+ \hat{I}_s^- + \hat{I}_r^- \hat{I}_s^+\right) + \sum_{p,q,r,s} \omega_{pqrs,xy}^{(\mathrm{eff})} \left(\hat{I}_p^+ \hat{I}_q^+ \hat{I}_r^- \hat{I}_s^- + \hat{I}_p^- \hat{I}_q^- \hat{I}_r^+ \hat{I}_s^+\right) \end{aligned} \tag{10}$$

Detailed expressions for the various effective coupling frequencies in Eqs. (9) and (10) can be found in the Supplementary Information or for the second-order terms in (Brunner et al., 1990b).

The second-order Hamiltonian is a three-spin zero-quantum type Hamiltonian with an additional $\hat{I}_z$ term as the third spin
operator. Time evolution under such a zero-quantum Hamiltonian does not lead to line splitting but a line shift as has been shown for the rank-one part of the $J$ coupling (Andrew and Farnell, 1968) which also has a purely zero-quantum Hamiltonian (Mehring, 1983). This fact has also been mentioned in a recent paper about homonuclear $J$ decoupling in solids (Moutzouri et al., 2020). As a consequence of this, the second-order Hamiltonian will not only lead to a line broadening but also a shift of the line that depends on the spinning frequency. In the presence of large chemical-shift differences, the second-order
Hamiltonian will be completely truncated and becomes invisible in the limit of weak coupling. The third-order Hamiltonian, however, contains also terms that lead to a splitting of the lines and are visible in the weak-coupling limit.

## 3 Numerical Simulations

All numerical simulations have been implemented using the spin-simulation environment GAMMA (Smith et al., 1994) which allows the use of arbitrary effective Hamiltonians. To characterize the line width in a homonuclear dipolar-coupled spin system,
two different approaches were used. In a first approach, the propagator of the time-dependent Hamiltonian over a full rotor cycle was calculated using time-slicing of the Hamiltonian. The effective Hamiltonian over the rotor cycle was then back calculated according to

$$\hat{\bar{\mathcal{H}}} = \frac{\ln(U(\tau_{\mathrm{r}}))}{i2\pi\tau_{\mathrm{r}}}. \tag{11}$$





Here, $\tau_r = 2\pi/\omega_r$ is the cycle time of the MAS rotation. Typically, one has to be careful that the eigenvalues of the Hamiltonian can be multivalued with a multiple of $\omega_r = \frac{2\pi}{\tau_r}$ and the correct solution is unknown. In the case of fast MAS, however, the eigenvalues of the effective Hamiltonian are typically much smaller than the spinning frequency and this multivalued solution of the logarithm poses no problems. The second approach uses the series expansion of the effective Hamiltonian based on Floquet theory as presented in the Theory Section. This allows us to compare spectra or properties of spectra as a function of the different levels of approximation as provided by Floquet theory. We diagonalize the effective Hamiltonian obtained from either method and calculate transition frequencies ($\omega_{ij}$) from the difference of the eigenvalues of the Hamiltonian in the eigenbase. Initial density operator (usually $\hat{F}_x = \sum_n \hat{I}_{nx}$) and the detection operator (usually $\hat{F}^- = \sum I_{nx} - i \sum_n I_{ny}$) are transformed into the eigenbase of the Hamiltonian and the off-diagonal elements are used to determine the intensity of the transitions ($I_{ij} = (\sigma_0)_{ij}(d_{ji})^*$). Here, $\sigma_0$ is the initial density operator and $d$ is the detection operator. This is basically a standard frequency-based spectrum calculation that allows us to reconstruct the spectrum with any desired resolution. Either spectra are calculated by binning the transition frequencies into a spectral range with a given frequency resolution or the intensities and the frequencies are directly used to calculate the $n^{th}$ moment of the line according to

$$M_n = \frac{\sum\limits_{i,j<i} \omega_{ij}^n I_{ij}}{\sum\limits_{i,j<i} I_{ij}} \tag{12}$$

The second method has a higher precision because the rounding due to the binning of the frequency values is avoided. Instead of the second moment ($M_2$), we use the equivalent FWHM of a Gaussian line with the same $M_2$ which is given by FWMH $= 2\sqrt{2\ln(2)M_2}$ as a measure of the line width (Mehring, 1983).

## 4 Results and Discussions

The spin systems are characterized by the coordinates of the spins $\mathbf{r}_i = (x_i, y_i, z_i)$ leading to the distances $r_{ij}$ and a set of Euler angles $(0, \theta_{ij}, \phi_{ij})$. The anisotropy of the dipolar coupling is then given by $\delta_{ij} = -2\frac{\mu_0}{4\pi}\frac{\gamma_i \gamma_j \hbar}{r_{ij}^3}$. We start with discussing a three-spin system since many characteristics of the residual line width can already be seen in this simple spin system. We assume the spin-pair (12) is aligned along the z axis defined by the static magnetic field. In a planar three-spin system, a single Euler angle is sufficient to describe the orientation of each coupling with respect to the coupling (12). Figure 1a shows the geometry of the three-spin system used in the simulations. In a second step we will go to a four spin system where the fourth spin can either be in or out of the plane spanned by the spins 1, 2, and 3 (see Fig. 1b). The details of each spin system are always given in the figure captions.

### 4.1 Three-Spin System without Chemical-Shift Differences

For simplicity, we start out with a three-spin system without chemical-shift differences and coordinates $\mathbf{r}_1 = (0,0,0)$, $\mathbf{r}_2 = (0,0,1.75)$, $\mathbf{r}_3 = (3,0,0)$ (in units of Å) leading to distances and angles given by $r_{12} = 1.75\,\text{Å}$, $r_{13} = 3.00\,\text{Å}$, $r_{23} = 3.47\,\text{Å}$, $\theta_{13} = 90°$, $\theta_{23} = 120°$(see Fig. 1). This is a mimic for a strongly-coupled $CH_2$ spin system with one additional more distant



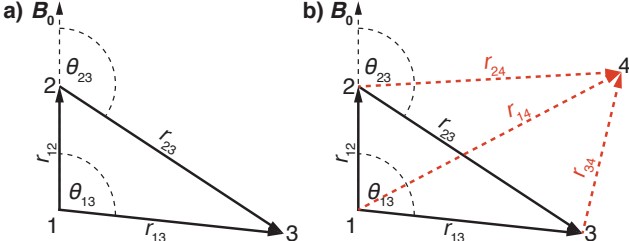

**Figure 1.** Schematic drawing of the a) three-spin and b) four-spin system used in numerical simulations indicating distances and relative orientations. The red dashed lines indicate out-of-plane vectors to the fourth spin.

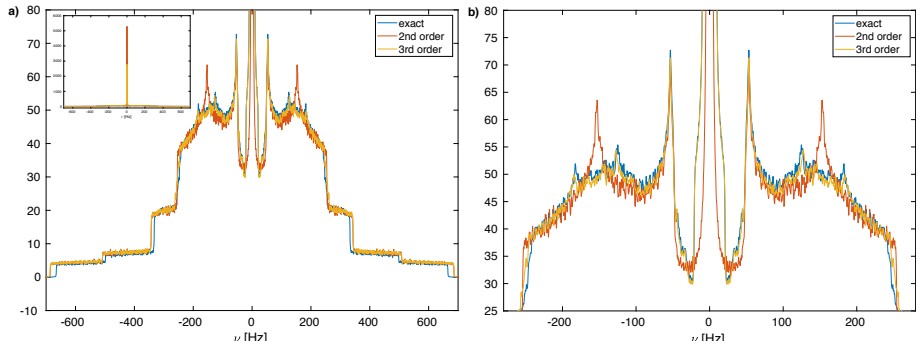

**Figure 2.** (a) Calculated MAS spectra at a spinning frequency of 50 kHz using different approximations of the effective Hamiltonian. All spectra were processed with an exponential line broadening of 1 Hz. The central very narrow peak is cut for a better display of the spectrum but the complete spectrum is shown as an inset in the upper left corner. The spin-system parameters are: $\delta_{12}/(2\pi) = -44826$ Hz, $\delta_{13}/(2\pi) = -8898$ Hz, $\delta_{23}/(2\pi) = -5750$ Hz, $\theta_{13} = 90°$, $\theta_{23} = 120°$. The spectral window was set to 2000 Hz with 20000 data points leading to a digital resolution of 0.1 Hz. One hundred thousand powder points were sampled according to the ZCW scheme (Cheng et al., 1973). (b) Expanded central region of the spectrum in (a). Here one can clearly see the differences between the second-order and the third-order approximation that is very close to the exact spectrum.

proton. The dipolar couplings in such a spin system are characterized by $\delta_{12}/(2\pi) = -44826$ Hz, $\delta_{13}/(2\pi) = -8898$ Hz,

135    $\delta_{23}/(2\pi) = -5750$ Hz. Figure 2a shows the calculated spectra for this spin system at an MAS frequency of 50 kHz using second-order, third-order and exact effective Hamiltonians. Note that the spectrum is cut off in the center at the top and the complete spectrum is shown as an inset in Fig. 2a. Spectra at other spinning frequencies ranging from 20 kHz to the currently experimentally inaccessible 1000 kHz can be found in the Supplementary Information (SI Fig. 1). An expanded view of the central part of the spectra is shown in Fig. 2b. The second-order Hamiltonian leads to a spectrum (red) that is already quite close

140    to the correct spectrum obtained from the numerically calculated effective Hamiltonian (blue) with some differences in features in the center part of the spectrum (see Fig. 2b). These differences are strongly reduced when we include the third-order terms (orange) which leads to a spectrum that has all the features and is very close to the exact one (blue) in the center of the line. The second-order and third-order spectrum, however, agree very well outside the central region. The fact, that the third-order





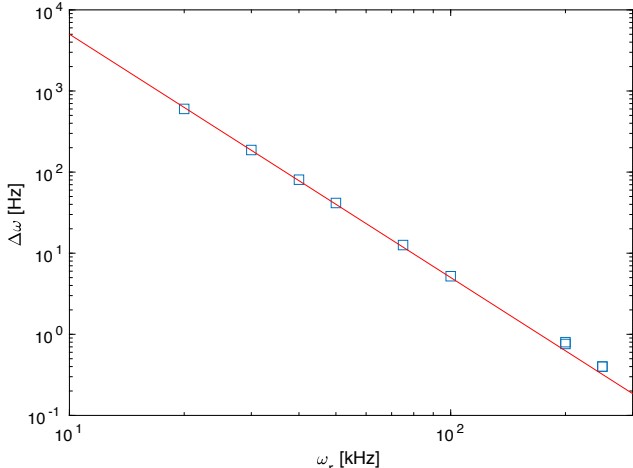

**Figure 3.** Difference in breadth of the simulated spectra between the third-order and the exact effective Hamiltonian as a function of the spinning frequency. The red line has a slope of -3 corresponding to a $\omega_r^{-3}$ dependence of the difference. This is the expected slope for a contribution by a fourth-order effective Hamiltonian.

Hamiltonian leads only to changes in the central part of the spectrum can be understood from the form of the Hamiltonian as shown in Eqs. (9) and (10). Since the two Hamiltonians do not commute, a large value for $\hat{\bar{\mathcal{H}}}^{(2)}$ will truncate $\hat{\bar{\mathcal{H}}}^{(3)}$ leading to negligible changes in the spectrum at larger offsets. In the center of the spectrum $\hat{\bar{\mathcal{H}}}^{(2)}$ is smaller and the influence of $\hat{\bar{\mathcal{H}}}^{(3)}$ is visible in the spectra.

However, there are clear differences in the line width of the exact spectrum and the spectrum based on second-order and third-order effective Hamiltonians which are clearly broader. This difference in the breadth of the powder line shape must be attributed to higher-order contributions to the effective Hamiltonian. Since calculating terms beyond the third-order term considered here is quite complex, we have investigated how the difference of the total breadth scales with spinning frequency. Figure 3 shows the difference in breadth in a double logarithmic plot for spinning frequencies ranging from 20 to 250 kHz. The data points lie on a line with slope -3 which indicates that the breadth depends on the spinning frequency with the inverse third power. This is a clear indication that the additional term that leads to the difference in breadth is a fourth-order effective Hamiltonian term that would scale with $\omega_r^{-3}$.

We can calculate the second moment of the powder lines as a function of the spinning frequency (see SI Fig. 1) as shown in Fig. 4. Above a spinning frequency of 50 kHz, there is a very good linear correlation between the line width and the inverse spinning frequency ($\omega_r^{-1}$) for all three sets of spectra. For second-order and third-order spectra, the correlation extends down to 20 kHz MAS frequency but for the full effective Hamiltonian, we observe a deviation from the linear correlation. This deviation towards narrower lines is most likely due to the fourth-order effective Hamiltonian as discussed above. This deviation becomes important for spinning frequencies below 50 kHz where the spinning frequency is in the order of the biggest dipolar coupling used in the model spin system. It is well known that perturbation expansions such as the average Hamiltonian or the van Vleck





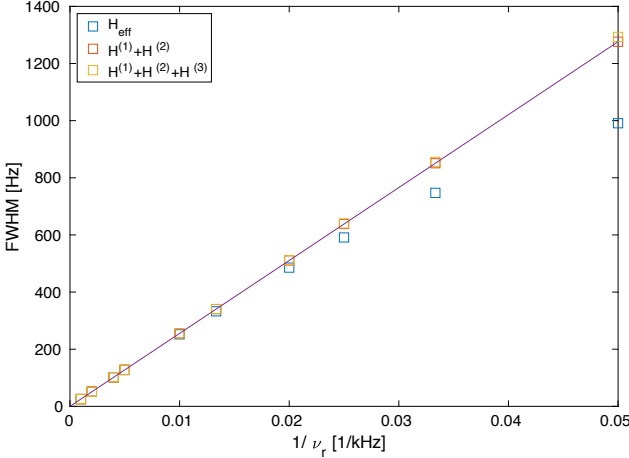

**Figure 4.** Line width (full-width half max) of a Gaussian line with the same second moment as the powder line shape shown in the SI Fig. 1 (FWMH $= 2\sqrt{2\ln(2)M_2}$). The line width obtained from second-order and third-order effective Hamiltonian correlates almost perfectly linear with $\omega_r^{-1}$ while the line width obtained from the exact effective Hamiltonian shows some deviations which becomes more prominent below 50 kHz. This deviation (narrower line width) is the contribution of the fourth-order term as discussed above (see Fig. 3).

expansion converge slowly in this regime (Blanes et al., 2009). It is interesting to note that the line widths for the second-order (red) and the third-order (yellow) spectra agree very well over the full range of spinning frequencies.

165    Of course, the second moment calculated over the complete spectrum including the side bands is preserved under MAS and independent of the spinning frequency (Lowe, 1959). Since we are interested in the line width of the center band, the second moment is calculated only over the center band of the line and we see a decrease in the second moment as a function of the spinning frequency and a corresponding decrease in the Gaussian line width.

## 4.2    Three-Spin System with Chemical-Shift Differences

170    Introducing chemical shifts makes the analysis of the spectra in terms of second moments and line widths more complex. This is due to the fact that we are now interested in second moments of the different lines that may overlap with each other or overlap with combination lines that are possible in strongly-coupled spin systems. Figure 5 shows the simulated proton spectra at 50 kHz MAS frequency for two different sets of chemical shifts. Spectra at different spinning frequencies (between 50 and 500 kHz) can be found in the SI Fig. 2. Slower spinning frequencies are difficult to analyze due to overlapping lines. Below the

175    spectra the regions of the various transition frequencies are marked by a black line. One can see that there are single-quantum transitions and combination lines with the combination lines having much lower intensity than the single-quantum transitions. Calculating the second moment over the complete line and correcting for the contributions by the isotropic chemical shifts confirms that the second moment is preserved as one would expect. However, this is not the quantity of interest since we are interested in the individual line width. We, therefore, calculate the first and second moment of the individual one-quantum lines

180    by restricting the calculations to the relevant areas of the spectrum indicated in the Figure by black lines.



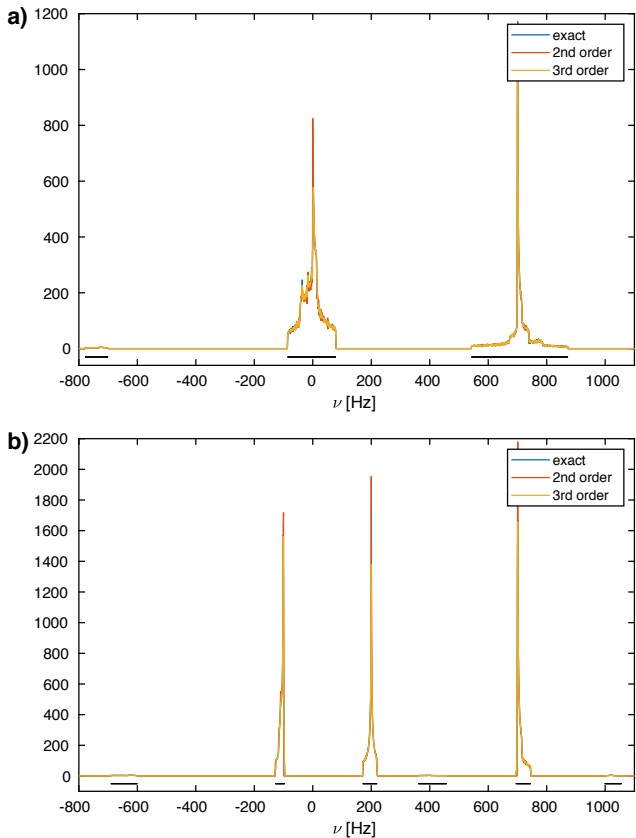

**Figure 5.** Calculated MAS spectra of a three-spin system at an MAS frequency of 50 kHz. The spin-system parameters are: $\delta_{12}/(2\pi) = -44826$ Hz, $\delta_{13}/(2\pi) = -8898$ Hz, $\delta_{23}/(2\pi) = -5750$ Hz, $\theta_{13} = 90°$, $\theta_{23} = 120°$. The spectral window was set to 20 kHz with 20000 data points leading to a digital resolution of 1 Hz. One hundred thousand powder points were sampled according to the ZCW scheme (Cheng et al., 1973). The chemical shifts were chosen to be a) $\delta_1 = \delta_2 = 0$ ppm, $\delta_3 = 0.7$ ppm and b) $\delta_1 = $ -0.1 ppm, $\delta_2 = 0.2$ ppm, $\delta_3 = 0.7$ ppm at a Larmor frequency of 1 GHz. The black line under the spectra indicates regions of transitions.

Calculating the transition-selective moments as a function of the MAS frequency shows that the first moment and the second moment are spinning-frequency dependent. The deviation of the first moment from the chemical shift is shown in Figs. 6a and b. One can clearly see that the differences between the different approximations (second order, third order and exact effective Hamiltonian) are quite small and that one can observe changes in the line position as a function of the spinning frequency. These changes in the line position will limit the accuracy of the chemical shift determination in proton spectra and will be discussed in more detail in Section 5. The dependence of the line width on the spinning frequency is shown in Figs. 6c-f for chemical shifts $\delta_1 = \delta_2 = 0$ ppm (squares), $\delta_3 = 0.7$ ppm (circles) (left column) and $\delta_1 = $ -0.1 ppm (squares), $\delta_2 = 0.2$ ppm (circles), $\delta_3 = 0.7$ ppm (triangles) (right column) as a function of $\nu_r^{-1}$ (middle row) and $\nu_r^{-2}$ (lower row). One can clearly see that for three-spin systems with chemical-shift differences the very clear $\nu_r^{-1}$ dependence of the three-spin systems without





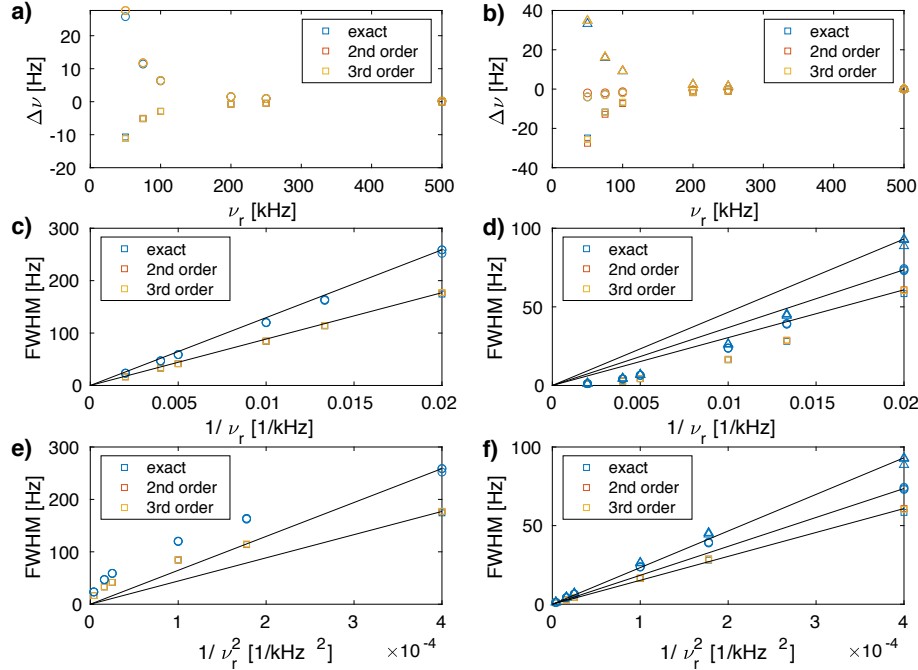

**Figure 6.** (a, b) Plot of the deviation of the calculated chemical shift (first moment of the line) from the theoretical chemical shift. Plot of the line width (FWHM) as a function of (c, d) $\nu_\mathrm{r}^{-1}$ and (e, f) $\nu_\mathrm{r}^{-2}$. The spin-system parameters are: $\delta_{12}/(2\pi) = -44826$ Hz, $\delta_{13}/(2\pi) = -8898$ Hz, $\delta_{23}/(2\pi) = -5750$ Hz, $\theta_{13} = 90°$, $\theta_{23} = 120°$. The chemical shifts were chosen to be (a,c,e) $\delta_1 = \delta_2 = 0$ ppm (squares), $\delta_3 = 0.7$ ppm (circles) and (b,d,f) $\delta_1 = -0.1$ ppm (squares), $\delta_2 = 0.2$ ppm (circles), $\delta_3 = 0.7$ ppm (triangles) at a Larmor frequency of 1 GHz.

chemical-shift differences does not hold anymore. For a strongly coupled spin pair with a small chemical shift difference and an additional coupling to a third spin with a strongly different chemical shift (Fig. 6c and d), the dependence of the line width on the spinning frequency is still quite close to $\nu_\mathrm{r}^{-1}$. For a three-spin system with three distinct chemical shifts, the MAS dependence is closer to $\nu_\mathrm{r}^{-2}$. In general, the dependence on the MAS frequency is somewhere between $\nu_\mathrm{r}^{-1}$ and $\nu_\mathrm{r}^{-2}$ and depends on the exact selection of the chemical shifts. We believe that this change in spinning-frequency dependence is a consequence of additional truncation of the second-order effective Hamiltonian by the chemical shifts. The truncation explains the experimentally observed deviation from the $\nu_\mathrm{r}^{-1}$ dependence of the homogeneous line width (Nishiyama, 2016; Sternberg et al., 2018; Penzel et al., 2019; Schledorn et al., 2020).

### 4.3 Larger Spin System without Chemical-Shift Differences

Figure 1b shows the geometry of the four-spin system where spin 4 can be out of the plane spanned by spins 1, 2, and 3. Therefore, the directions of $r_{14}$, $r_{24}$, and $r_{34}$ are defined by sets of two Euler angles each $(0, \theta_{ij}, \phi_{ij})$. For simplicity, we will show simulations for a four-spin system where all four spins are in a plane, i.e., $\phi_{ij} = 0$. Simulations on spin systems with out-of-plane spins showed exactly the same behavior. The details of the spin system are given in the figure captions.

MAGNETIC
RESONANCE
Open Access Discussions

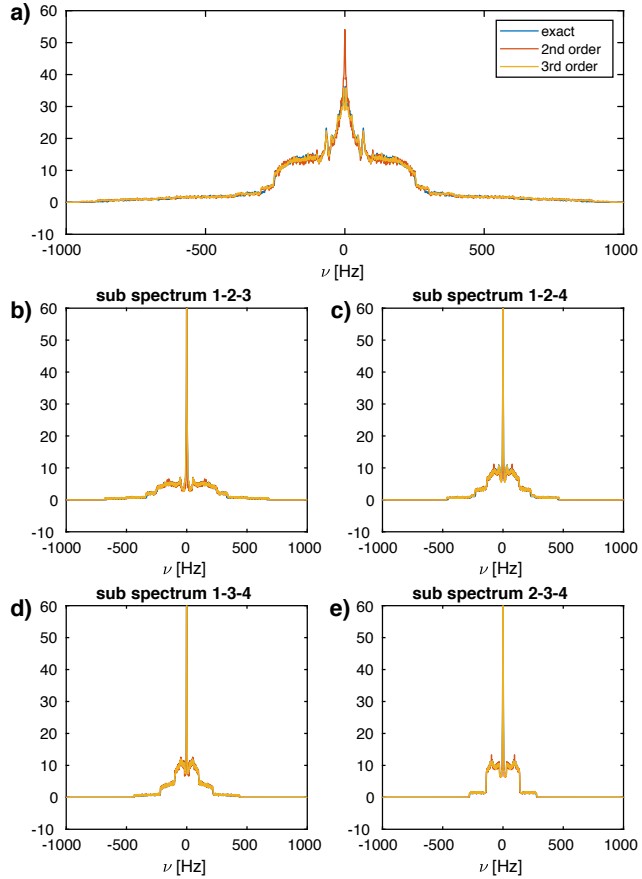

**Figure 7.** a) Calculated MAS spectra for a four-spin system at a spinning frequency of 50 kHz using different approximations of the effective Hamiltonian. All spectra were processed with an exponential line broadening of 1 Hz. The spin-system parameters are: $r_1 = (0,0,0)$Å, $r_2 = (0,0,1.75)$Å, $r_3 = (3.0,0,0)$Å, $r_4 = (3.5,0.0,2.0)$Å, $\delta_{12}/(2\pi) = -44826$ Hz, $\delta_{13}/(2\pi) = -8898$ Hz, $\delta_{14}/(2\pi) = -3667$ Hz, $\delta_{23}/(2\pi) = -5734$ Hz, $\delta_{24}/(2\pi) = -5561$ Hz, $\delta_{34}/(2\pi) = -27420$ Hz. The Euler angles can be calculated from the coordinates given above. The spectral window was set to 2000 Hz with 20000 datapoints leading to a digital resolution of 0.1 Hz. Ten thousand powder points were sampled according to the ZCW scheme (Cheng et al., 1973). (b)-(e) shows the four three-spin sub spectra that comprise the complete four-spin spectrum shown in (a).

Figure 7a shows spectra for a four-spin system using second-order, third-order, and exact effective Hamiltonians. All three spectra are virtually identical except for the central sharp peak that is significantly higher in the second-order approximations. In addition, the narrower powder line shapes for the exact effective Hamiltonian are observed as in the three-spin simulations. The second moment of the powder line shape in the four-spin systems scales as a function of the MAS frequency in the same way as for the three-spin systems (see SI Figs. 3 and 4) almost perfectly with $\omega_r^{-1}$ with some small deviations visible at spinning frequencies below 75 kHz. This behavior is exactly the same as in the three-spin case. If we calculate the second moment of the four-spin system and compare it to the sum of the four three-spin sub systems, we find $M_2^{(1234)} = \frac{3}{4}(M_2^{(123)} +$





$M_2^{(124)} + M_2^{(134)} + M_2^{(234)}$) to within an error of less than 1% for all spinning frequencies. This allows us to calculate the line width of a multi-spin system based on the three-spin sub systems as along as the second-order contribution is dominating the residual line width under MAS. The same behavior is observed for five-spin systems that can de decomposed in four-spin or three-spin sub systems for the calculation of the second moment as long as the second-order contribution dominates the line width (see SI Figs. 5 and 6). Again, the deviation of the calculated second moments is less than 1%.

## 5  Experimental Data

We also tried to experimentally characterize the line position changes associated with the time evolution under the second-order Hamiltonian as expected from Eq. (9). As shown in Figure 6a, such effects are only on the order of several Hz and, therefore, hard to extract from experimental proton-detected solid-state NMR spectra. To enhance proton resolution to a degree that makes the observation of such small effects possible, we recorded $^1$H MAS spectra of the crystalline compound ortho-
phospho-L-serine previously studied by solid-state NMR (Duma et al., 2008; Potrzebowski et al., 2003; Iuga and Brunner, 2004) at MAS frequencies ranging from 70 to 158 kHz and extracted the apparent isotropic chemical shifts of the methylene group $CH_2$ protons and the $C\alpha H$ proton by line-shape deconvolution of each spectrum using DMFIT4 (Massiot et al., 2002). This becomes possible due to decreasing coherent line broadening contributions to the proton line width at fast MAS (Sternberg et al., 2018; Malär et al., 2019). The chemical-shift deviations for the three protons from the values obtained at 158 kHz are
given in Figure 8 and indeed reveal the theoretically predicted dependence on the MAS spinning frequency. We also simulated the line-shift changes by using a simplified three-spin system with parameters relying on the crystal structure (CSD entry SERPOP03). And indeed, the experimentally observed trends can be reproduced reasonably well by these simple three-spin simulations taking second-order effects into account. However, an even more accurate experimental determination of such an effect is still to some extent limited by the residual broadening of proton resonances in this crystalline compound (see Figure
8c).

## 6  Conclusions

We have shown through numerical simulations using various orders of effective Hamiltonians that second-order dipolar contributions under MAS lead to a MAS dependence of the line position and dominate the residual line broadening in dipolar-coupled homonuclear spin systems. Third-order terms do not play a significant role for the residual line width but change the line shape close to the center of the line. Fourth-order terms were not explicitly calculated but were shown to be contributing to the line
width at MAS frequencies below 50 kHz in strongly-coupled proton spin systems. Without chemical-shift differences we observe a clear $\omega_r^{-1}$ dependence of the residual line width under MAS in three-spin as well as larger spin systems above 50 kHz MAS. The implementation of chemical-shift differences leads to a change of this spinning-frequency dependence in three-spin systems. The clear $\omega_r^{-1}$ dependence changes to a spinning-frequency dependence somewhere between $\omega_r^{-1}$ and $\omega_r^{-2}$ depending
on the details of the involved chemical shifts. Looking at larger spin systems with chemical-shift differences is more complex

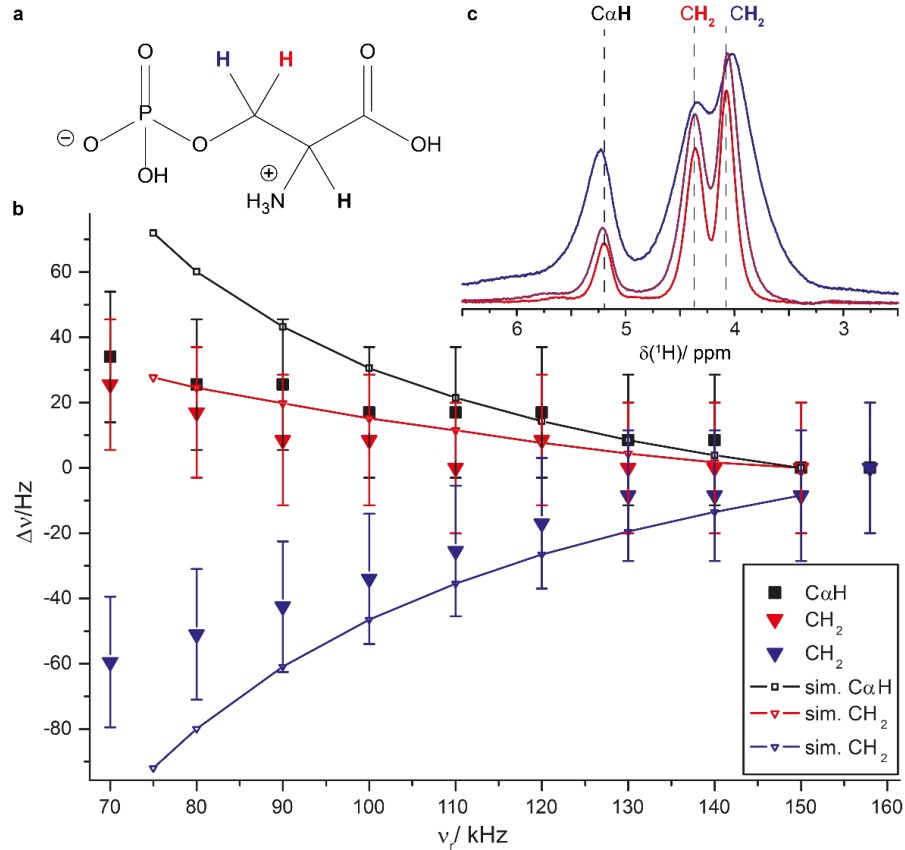

**Figure 8.** MAS dependence of $^1$H line positions in ortho-phospho-L-serine (for the chemical structure see a). b The filled symbols indicate experimental data with a constant error of 20 Hz. The two $CH_2$ protons are shown in red (high ppm peak) and blue (low ppm peak), the $C\alpha$H proton is shown in black. The simulations were based on a three-spin system with $r_{12} = 1.60$ Å, $r_{23} = r_{13} = 2.35$ Å, and $\theta(1,2,3) = \theta(2,1,3) = 70°$. Three $^1$H spectra recorded with MAS frequencies of 70 kHz (blue), 120 kHz (purple) and 158 kHz (red) are shown in c. The dashed lines indicate the isotropic $^1$H chemical shift at 158 kHz MAS. The experiments were performed at a static magnetic field of 20 T.

using the approach used here since the separation of the lines becomes more difficult due to the larger number of combination lines. We are currently working on this problem that is beyond the scope of this manuscript.

*Data availability.* All data will be made available through the ETH library data services before publication of the final version.

*Supplement.* The supplement related to this article is available online at: https://doi.org/10.5194/mr-0-1-2021-supplement.



*Author contributions.* BM and ME designed the research. MC and ME did the theoretical Floquet calculations. ME carried out the numerical simulations. AM and TW conducted the experiments. All authors discussed the results and contributed to writing the manuscript.

*Competing interests.* The authors declare that they have no conflict of interest.

*Acknowledgements.* We would like to thank Dr. Ago Samoson and his team for providing the probe used in the experimental measurements.

*Financial support.* This research has been supported by the Schweizerischer Nationalfonds zur Förderung der Wissenschaftlichen Forschung
(grant no. 200020_188988).





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
