# Peer review of "Residual Dipolar Linewidth in Magic-Angle Spinning Proton Solid-State NMR"

_Magnetic Resonance, 2021_

## Author Response (AR1)

Response to Reviewer 1:

Thank you for the positive comments and the detailed reading of the manuscript.

Q: Fig. 4 caption, 3rd line: deviations which become

A: corrected

Q: Fig. 6 explain the straight lines in the caption

A: The lines are a guide to the eye to show the linear correlation. They go through the
… point (0,0) and the data point for the lowest MAS frequency. We have added the following
… sentences to Figure caption 6: "The straight lines go through (0,0) and the line width at
… the slowest spinning frequency. They are meant as a guide to the eye for the linear
… correlation."

Q: Fig. 8 avoid repeat of bue and red in c to avoid confusion with colours used in b: how
… about green and orange instead?

A: We have changed the coloring scheme to avoid confusion.

Q: Can the SI use S1 etc labels for page numbers, sections, Tables and Figures.

A: We have changed the labelling of the SI to include the "S" for figures, pages and
… tables.

Q: SI Figure 2: line 4, space between set and between

A: corrected as well as another error in this caption.

Response to Reviewer 2:

Q: ABMS, like proton-proton dipolar interactions, yield a MAS dependent line width.  How
… do the authors differentiate between the two effects.  The same applies to the MAS
… induced frequency shift (see e.g. Alla and Lippmaa, Chem. Phys. Lett. 1982, 87, 30-33;
… Samoson et al. Solid State Nucl. Magn. Res. 2001, 20, 130-136).  While reading through
… the paper, one gets the impression that dipole-dipole interactions dominate the proton
… line width.  The abstract / title should be modified into something like "Residual line
… width resulting from proton dipolar interactions in Magic-Angle Spinning Solid-State NMR"
… in order to avoid confusions.  To differentiate, some experimental data on the field
… dependence of the MAS dependent line width would be highly appreciated.

A: We agree that the title should include the fact that we exclusively discuss broadening
… from dipolar contributions. Therefore, we have changed the title to "Residual Dipolar
… Linewidth in Magic-Angle Spinning Proton Solid-State NMR". We have also added dipolar in
… the abstract in two locations. The influence of ABMS on line position and line width has
… been proven to be elusive in the literature. The way we understand this effect is that
… ABMS originates from the shape and finite dimensions of the crystallites present in the
… powder sample. MAS averages ABMS contributions from the isotropic susceptibility but not
… ABMS effects originating from the anisotropic part of the susceptibility. The reason for
… this is the fact that they can be described as the product of two second-rank tensors
… that generate rank-0-2-4 components. Therefore we expect a line shift due to anisotropic
… ABMS contributions, rank-2 that are averaged out and rank-4 that are scaled but not fully
… averaged out (see Alla and Lippmaa, Chem. Phys. Lett. 1982, 87, 30-33 section 3).
… However, the ABMS line width should scale with $P_4(\cos(\theta_m))$ and should, in
… principle, be MAS independent as should be the shift. They are both $B_0$ field dependent
… as is experimentally shown in Samoson et al. Solid State Nucl. Magn. Res. 2001, 20,
… 130-136. If our understanding of the ABMS shifts is wrong, we would appreciate more
… pointers where we misunderstood the literature.

We have not included experimental data since we are mostly interested in the theoretical
… underlying mechanism of the changes in scaling of the MAS line width. We reference

31… several papers that discuss this experimental finding in the introduction (around line
… 40): "Experimental observations of the residual homogeneous line width as a function of
… spinning frequency show that it can often be approximated by a linear correlation with
… the inverse of the spinning frequency with some deviation that indicate a partial inverse
… quadratic dependence (Nishiyama, 2016; Sternberg et al., 2018; Penzel et al., 2019;
… Schledorn et al., 2020). This has been attributed to third-order contributions to the
… effective Hamiltonian or to chemical-shift effects (Sternberg et al., 2018; Moutzouri et
… al., 2020)." We hope that this is sufficient.

Q: In the manuscript, the authors do a great job in discussing the MAS dependent 1H line
… width.  Unfortunately, the contribution to the signal that is hidden in the Pake like
… pattern in the base line is not quantified.  Can the authors give an estimate how the
… intensity is changed with MAS frequency using second-order Hamiltonian arguments ?  This
… would be extremely interesting, since most solid-state NMR experiments are sensitivity-
… and not resolution-limited.  The manuscript is very similar to a recent paper by Xue et
… al. (J. Phys. Chem. C 2018, 122, 16437).  The authors should discuss theoretical versus
… computational approaches to yield an understanding of proton resonances in the
… solid-state. In the presented approach, the geometry is restricted to very few angles and
… distances.  Is it possible to derive general laws if only a few spins are considered ?

A: The line width is easy to assess via an expansion of the moments but of course there
… are many distributions that produce the same (second) moment and an easy guess of the
… line height is not possible. Of course in the limit of many spins, we might get a
… Gaussian line and then we could predict the line height. However, to what extend this is
… true, we do not know yet and is part of our current research efforts.
The appendix (and also references cited in the paper) give analytical expressions for the
… second- and third-order Hamiltonian. Based on these expressions, one can calculate
… analytical solutions of the line width. However, the expressions become very complex
… functions of the distances and relative orientations of the couplings and it is not easy
… to get much insight from them beyond the form of the spin operators. This is the reason
… that we opted for a numerical implementation of the effective Hamiltonians. The strength
… of this method is that we can distinguish which order of the effective Hamiltonian
… contributes which is not accessible from a purely numerical simulation as in the work by
… Xue.

Q: While reading the paper, one gets the impression that a proton line can be infinitely
… narrow if only the MAS frequency is high enough. The authors should add an additional
… term to their equations which summarizes the contributions to line width that are not
… affected by MAS.  At which MAS frequency does the 1/omega_r dependence break down ?

A: The homonuclear dipolar part of the line width that is discussed in the paper has no
… spinning-frequency independent term. Of course, other homogeneous parts of the line
… width, e.g., relaxation terms or chemical exchange and inhomogeneous terms, e.g., sample
… inhomogeneity or ABMS shifts might broaden the line in addition to the dipolar line width
… as discussed in the introduction (around line 25). It depends on the relative magnitude
… of the various contributions where the 1/omega_r dependence breaks down and no general
… rule can be given. Especially, sample inhomogeneity and ABMS contributions can vary over
… a large range of values.

Q: I am missing a paragraph in which the theoretical considerations are compared with
… experimental data, and correlate theory and experiment.  The author did this for the MAS
… induced frequency shift shown in fig 8 which is very nice.  However, I am missing a
… discussion on the line width.  The data for phospho-serine exist (at least in the MAS
… range 70-160 kHz), and it should be straight forward to read out the line width after a
… line shape fit.

A: The isotropic line shift can be compared quite well because it looks like the dipolar
… contribution to the isotropic line shift is the dominating one. As discussed above, there
… are many contributions to the line broadening and we do not claim that we can predict the
… line width from one simple three-spin simulation. In addition, the focus of the paper is
… on understanding the spinning-frequency dependence of the line broadening and not
… predicting the line broadening. Therefore, we do not want to make comparison to

44… experimental data of the line width but give references to published data that illustrate
… the spinning frequency dependence.

---

## Author Response (AR2)

I would like to also consider the following points:
Q: * in the experimental data on phospho-serine, you show a MAS dependency of line widths
and positions. Peak positions
are, in general, temperature dependent. As a change of the MAS frequency generally leads
to a change of the effective
temperature, the question is whether temperature effects contribute to the observed
shifts. Please specify how the
temperature was regulated, and comment on the possibility of temperature effects.

A: The selected lines in phospho serine (CH2 and Halpha) do not show a measurable
temperature dependence which is the erason why we used them to check this effect. We have
added a sentence to the manuscript reading: "The resonances of ortho-phospho-L-serine
reported in the experimental section (the methylene CH2 and the CαH protons) do not show a
measurable temperature dependence of the chemcial-shift values as described recently
(Malär et al., 2021)." The reference has the temperature-dependence data.

Q: * in Figure 4, I would find it useful to add additional ticks (maybe above the graph)
that show the MAS frequency,
rather than its inverse; e.g. label 50 kHz at the 0.02 tick etc. This is arguably a
cosmetic operation of little
importance, but most people think in nu_r rather than in tau_r.

A: We have modified Figure 4 as requested with a \nu_r axis on the top.